# An Optimised MS-Based Versatile Untargeted Metabolomics Protocol

**Cátia F. Marques**  **and Gonçalo C. Justino \*** 

Centro de Química Estrutural–Institute of Molecular Sciences, Instituto Superior Técnico, Universidade de Lisboa, 1049-001 Lisboa, Portugal
\* Correspondence: goncalo.justino@tecnico.ulisboa.pt

**Abstract:** Untargeted metabolomics approaches require complex samples containing the endogenous metabolites of a biological system. Here, we describe a set of protocols that can be applied to various types of samples, including prokaryotic and eukaryotic cells, as well as animal and human samples. Following a single extraction step, samples are analysed using different chromatographic conditions coupled to high-resolution mass spectrometry. Quantification of metabolite changes between samples is performed without internal standards, using peak areas from extracted ion chromatograms for statistical analysis. Bioinformatics annotation of the results allows a pathway- and process-oriented analysis across biological sample conditions, allowing a complete pathway interrogation.

**Keywords:** metabolomics; mouse; human; yeast; bacteria; culture cells; tissues; plasma; urine; faeces; ESI-UHPLC-MS

## 1. Introduction

The large-scale study of metabolites (substrates and products of metabolism that drive essential cellular functions) is known as metabolomics and takes advantage of the innovative developments in analytical and bioinformatics tools [1]. Metabolomics comprises pharmacometabolomics, an emerging branch that evaluates samples after exposure to a drug or another xenobiotic in order to establish possible models of treatment outcomes (e.g., efficacy or toxicity) and understand mechanisms of action. Pharmacometabolomics is a promising tool for personalized medicine, biomarker discovery, toxicology, and drug development, since it allows the elucidation of exposure-related alterations in biological pathways and their implications in downstream physiological and pathological conditions [2–4]. Metabolomics studies can also be employed to understand the pathophysiological alterations that occur in biological systems. Infection and disease, for example, affect organisms in multiple ways, leading to changes in various metabolic pathways. By combining a metabolomics analytical approach with a pathway-oriented bioinformatics annotation, it is possible to obtain a clear picture of the dysregulated pathways in each case, allowing a deeper understanding of the molecular basis of each disease [5]. In parallel, untargeted metabolomic studies of patients' samples have the potential to unravel novel candidate biomarkers that can ultimately translate into a faster disease diagnosis and management [1,6].

Metabolomics studies typically start with sample collection (e.g., cell extract, tissue, plasma, serum, and urine), metabolism arrest, and metabolite extraction for further separation, analysis, and quantification. Two methodologies can be considered for metabolite recovery and identification: targeted and untargeted (global) approaches. In a targeted metabolomics analysis, well-defined, and well-characterized metabolites are specifically investigated using specific and optimized extraction methods. In contrast, untargeted metabolomics approaches aim to measure all metabolites extracted from a sample, both known and unknown, creating a complex dataset that is analysed by computational tools and used to establish a relation between metabolites and metabolic pathways [1,7].

### 1.1. Metabolomics by Mass Spectrometry

Despite the current use of nuclear magnetic resonance (NMR) spectroscopy for the study of metabolomic profiles, high-resolution mass spectrometry (HRMS) is a more sensitive, selective, and versatile tool. Mass spectrometers are usually coupled to chromatography interfaces, allowing the separation of the metabolites via gas chromatography (GC-MS), high-performance liquid chromatography (HPLC-MS), or ultra-high-performance liquid chromatography (UHPLC-MS). Different column chemistries and mobile phases are also routinely used to improve the amount of identified metabolites [1,8,9].

MS-based metabolomics characterizes metabolites based on their *m/z* value, retention time, and predicted molecular formula. Tandem mass spectrometry (MS/MS) experiments allow structural elucidation based on the fragmentation patterns of metabolites. In addition, the use of a semi-targeted analysis, multiple reaction monitoring (MRM), is also helpful in the identification of metabolites of interest by searching for loss of specific fragments [8–10].

### 1.2. Development of the Protocol

We present here a detailed working protocol for characterizing the metabolome from various types of samples. Initially aimed at the characterization of the metabolomic profile from various mouse organs and tissues, within a medicinal chemistry framework, targeting the analysis of drug action, this protocol has evolved to accommodate many more sample types, from various species, addressing the biochemical characterization of a system's response to various stimuli. This system-wide approach is well-suited for exploratory goals, where one aims to identify biological pathways of potential relevance to the biological conditions assayed.

We start by addressing the issue of metabolome extraction from various samples, and how to properly arrest cellular metabolism. Then, we proceeded by optimizing the extraction protocols, to cover the widest possible set of metabolites. Finally, we address the chromatographic resolution and mass spectrometry-based identification of the metabolites present in the samples.

### 1.3. Advantages of the Method

The holistic approach behind untargeted metabolomics requires analytical methods capable of delivering large datasets of information per sample for further analysis. Within this paradigm, one's goal must be to extract the largest possible number of metabolites from each sample in a minimal number of steps and to analyse them through various chromatographic settings as required. The extractions procedures adopted follow this approach. The goal is not to split the metabolome in various fractions, but rather to fully interrogate it in a single step, affording large datasets that can be mined for information from different perspectives.

### 1.4. Current Limitations

Despite the advantages of MS-metabolomics, this approach also has some limitations. First, spectral annotation, i.e., identification of the metabolite that generated each observed ion, requires extensive databases. The immediate experimental results of MS-based untargeted metabolomics experiments are lists of *m/z* ratios, peak intensities, and retention times for each ionic peak identified using the mass spectrometer. While extensive databases are available for some species, information is scarce for other species. Second, data-dependent acquisition strategies are routinely used, where MS/MS information is acquired only for the most intense ions in each instant, following a data-dependent acquisition strategy. Thus, many of the ions observed in a full MS scan do not have any MS/MS information for identification confirmation. Third, a global (untargeted) approach cannot be used for absolute quantification, but only for relative quantification between samples. To address absolute quantification goals, a targeted approach must be considered.

Another aspect that must be properly evaluated is the occurrence of severe matrix effects causing signal suppression at the MS stage. If this is suspected, addition of a deuterated internal standard (IS) to the samples can be used. The IS peak area from spiked samples must be equivalent to the area determined by analysing the pure compound. To Remove matrix effects requires, among others, fine tuning the extraction protocol or using other protocols, optimized specifically for each biological sample.

## 2. Experimental Design

This tutorial presents a set of protocols that can be used in the preparation of a variety of samples, including prokaryotic and eukaryotic cells, animal samples, and human samples.

To minimize potential changes in sample composition before metabolite extraction, all steps should be performed on ice. While some sample preparation protocols rely on sample drying before analysis, we prefer not to. We have often observed that solvent evaporation leads to the formation of viscous samples that are not easily analysed. Instead, we rely on the use of adequate solvent mixtures and extraction procedures, coupled to ultra-high-resolution high-sensitivity mass spectrometry, to analyse all species present in the samples, even at very low (sub-micromolar) concentrations. Upon extraction, samples were promptly analysed or stored at $-80\,^{\circ}$C before LC/MS analysis.

Metabolites extracted with the protocols described below are not suitable for lipidomics analysis, as the extraction protocols are not designed for highly apolar lipid molecules.

For metabolomics assays using cells (prokaryotic or eukaryotic), we report a minimal or usual number of cells harvested in each case. It is possible to work with larger cell numbers, although in these cases, care must be taken at the MS step, as samples richer in cells are also richer in potential contaminants. Lower cell numbers can also be used but must be tested. As a rule of thumb, a minimal cell number is one that, when cells are pelleted in a microtube, originates a visible cell pellet.

The protocols presented below are schematized in Figures 1–5 and are described in detail in the Detailed Procedure section (Supplementary Materials).

### 2.1. Bacteria

Various protocols are available for metabolite extraction from bacteria, ranging from simple osmotic shock to French press extraction, including smoother lysozyme-based methods, depending on whether bacteria are Gram-positive or -negative. We have tested various methods over time, to various degrees of extraction. More comprehensive reviews of bacteria metabolites extraction methods are available in the literature [11–13]. The best overall method is the freeze–thaw approach, using liquid nitrogen for a faster turnover, for both Gram-positive and -negative species (Figure 1). This method requires that a CFU vs. OD600 curve be obtained beforehand, to allow for the determining of the number of cells from OD readings.

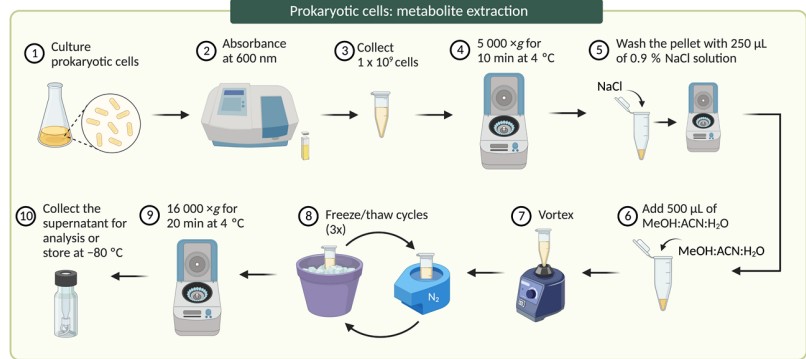

**Figure 1.** Scheme of metabolite extraction from prokaryotic cells. Image created with BioRender.com (https://www.biorender.com accessed on 17 May 2023).

## 2.2. Eukaryotic Cells

Two different approaches were optimised for metabolite extraction from eukaryotic cells. The first consists of successive freeze–thaw cycles using liquid nitrogen, and was tested on yeast cells and mammalian cells, both in suspension and detached cells. The second process allows metabolite extraction from plated cells, by direct scraping in the culture dish. Various methods are available in the literature, for both yeast and culture cells, with different specificities in mind [14–16].

Despite being the standard procedure for the detachment of adherence cells, allowing for cell counting, trypsinization is not suited for metabolomics approaches. During trypsinization, the cell metabolism is not immediately quenched, and trypsin can induce metabolite leaking during the whole process [17,18]. We have previously applied this protocol in order to extract metabolites from cultured cells [19].

## 2.3. Metabolite Extraction from Suspension Cells Using Freeze–Thaw Cycles

Cells are collected into a microtube and pelleted via centrifugation at $150\times g$ for 10 min at 4 °C. Upon washing with an iso-osmotic solvent, cells are resuspended in methanol–acetonitrile–water (2:2:1 *vol/vol/vol*) extraction mixture and submitted to three freeze–thaw cycles using liquid nitrogen. Metabolites are then collected in the supernatant and analysed using LC/MS. This protocol, optimized from the literature for a faster sample processing [14,15], is summarized in Figure 2.

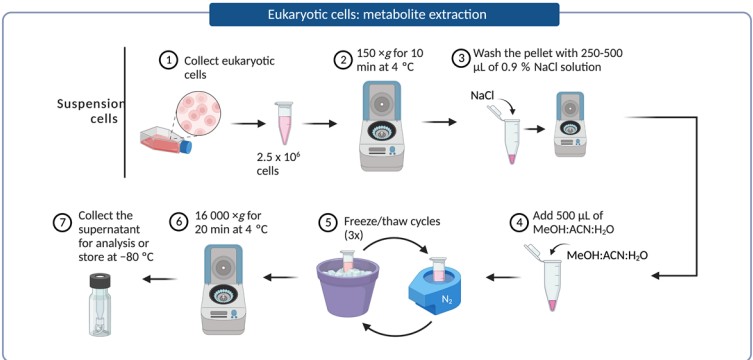

**Figure 2.** Scheme of metabolite extraction from suspension eukaryotic cells. Image created with BioRender.com (https://www.biorender.com accessed on 17 May 2023).

## 2.4. Metabolite Extraction from Adherent Cells by Cell Scraping

Upon saline washing, cell metabolism is arrested with cold methanol–water (4:1, *vol/vol*) solution [14]. Cells are scraped into this solution, which is collected for further processing. This protocol is depicted in Figure 3.

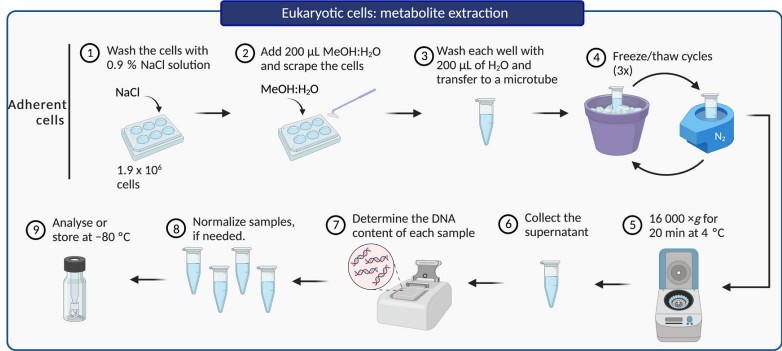

**Figure 3.** Scheme of metabolite extraction from adherent eukaryotic cells. Image created with BioRender.com (https://www.biorender.com accessed on 17 May 2023).

### 2.5. Mouse Samples

The preparation of samples for metabolomics analysis is the major bottleneck of this omics approach. Anaesthesia (e.g., continuous isoflurane, ketamine, and pentobarbital) and sacrifice (e.g., cervical dislocation, carbon dioxide, and isoflurane overdose) can lead to differences in metabolite expression, and it is difficult to decide which method is more adequate when compared with the other. According to Overmyer et al. [20], isoflurane anaesthesia leads to less metabolite variability in C57BL/6J mice than that experienced with pentobarbital and ketamine. To minimize animal stress and metabolite variability, we opted for anaesthetizing C57BL/6J mice with isoflurane and sacrificing them by cervical dislocation [19]. Tissues, biofluids, and faecal matter are collected and stored immediately in liquid nitrogen, followed by storage at −80 °C until processing. This protocol is summarized in Figure 4.

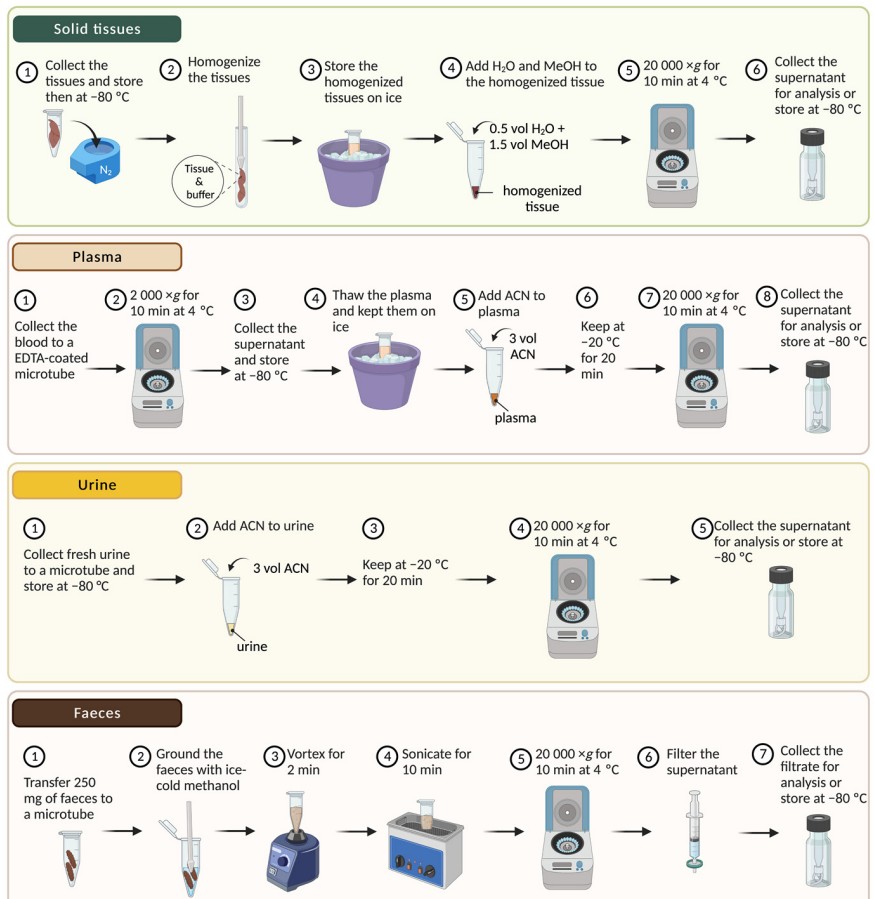

**Figure 4.** Scheme of metabolite extraction from mouse samples. Image created with BioRender.com (https://www.biorender.com accessed on 17 May 2023).

### 2.6. Mouse Tissue Sample Preparation

Many organs are characterized by their heterogeneity [21] (e.g., the liver has five different topographic lobes with different enzyme levels, and the kidney is characterised by different structures such as the medulla and the cortex). Thus, metabolite analysis of the different tissues can be performed considering each separate region or the entire organ. If organ spatial information is not necessary, tissues can be wholly homogenized in LC/MS water at 0.5–1 mg of tissue/mL.

Tissues are thawed on ice in order to minimise any possible enzyme activity and ground in ice-cold LC/MS water on a 3 mL Potter-Elvehjem homogenizer or a 1.5 mL microtube using a micro-pestle [22]. For liver, kidney, and intestine tissues, tissues were

homogenized at 1 mg of tissue/mL, while the brain, lung, hippocampus, and pre-frontal cortex were homogenized at 0.5 mg of tissue/mL. The choice between the homogenization type (Potter vs. micro-pestle), as well as the mass–volume concentration of each sample, is adjustable according to the tissue size.

Upon tissue homogenization, metabolites are extracted using a final mixture of 1.5:1.5 (*vol/vol*) water–methanol. The metabolites present in the supernatant are analysed by LC/MS.

### 2.7. Mouse Plasma Sample Preparation

According to the literature, the differences between plasma and serum samples include the levels of phospholipids (most likely due to the platelet phospholipase activity during serum clot formation [23]), amino acid content, and glucose, citrate, and pyruvate, among others [24–26].

We chose to use EDTA-treated plasma from mouse blood samples since plasma tubes can be stored on ice immediately after centrifugation, and also because EDTA can inhibit several enzymes and, by preventing coagulation, prevent clotting-associated enzyme release [25].

The protocol for metabolite extraction from plasma was adapted from Want [27]. A 50 μL sample is diluted with three volumes of ice-cold acetonitrile and kept at −20 °C for 20 min. Upon centrifugation at 20,000× $g$ for 10 min at 4 °C, 100 μL of the supernatant are collected for analysis by LC/MS.

### 2.8. Mouse Urine Sample Preparation

The protocol for metabolite extraction from urine was adapted from Want [27]. A 50 μL sample is diluted with three volumes of ice-cold acetonitrile and kept at −20 °C for 20 min. Upon centrifugation at 20,000× $g$ for 10 min at 4 °C, 100 μL of the upper fraction and 50 μL of the lower fraction are collected for analysis using LC/MS.

### 2.9. Mouse Faecal Matter Samples Preparation

The extraction of metabolites from faecal samples was adapted from Deda et al. [28]. Briefly, a 250 mg sample of faeces is ground with 1 mL ice-cold methanol, vortexed for 2 min, and sonicated for 10 min. After a 30 min centrifugation at 20,000× $g$ at 4 °C, the supernatant is filtered through a 0.22 μm polyethersulfone (PES) membrane, and the filtrate is analysed via LC/MS. Both dry faeces, collected from the cage floor, and wet faeces, collected from the animal intestines, can be used for metabolomics assays.

### 2.10. Human Samples

Despite the benefits of using plasma samples vs. serum samples described above, human blood samples are more easily available as serum and not as plasma, as the serum is easier to manipulate. Thus, sample preparation for human serum is included, following a methanol–acetonitrile–water extraction protocol [29], as depicted in Figure 5.

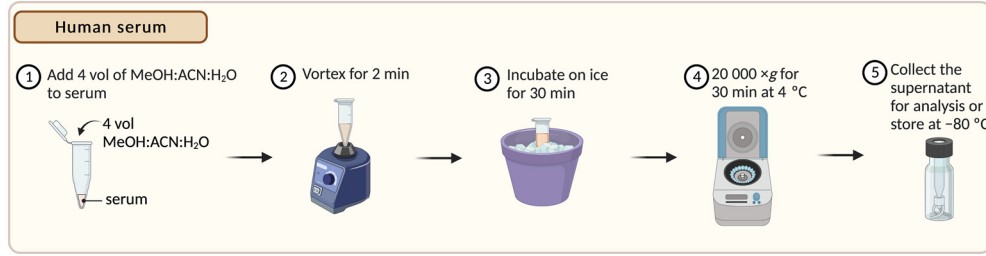

**Figure 5.** Scheme of metabolite extraction from human serum samples. Image created with BioRender.com (https://www.biorender.com accessed on 17 May 2023).

### 2.11. Controls

Adequate controls must be prepared for each case. For example, to analyse metabolome changes of a cell culture exposed to a drug, a control should be prepared as if it was a sample but replacing the drug solution with the same volume of the solvent used to prepare the drug aliquots. These controls are processed and analysed in the same way as samples, and, during data analysis, all samples are compared to these controls.

### 2.12. Data Processing

Upon analysis of all samples via mass spectrometry, in both positive and negative electrospray (ESI) modes, raw data are processed, analysed and interpreted. While some manufacturers have their own metabolomics data treatment software, open-source manufacturer-independent packages are available. From the available software, Mzmine [30,31], XCMS [32–34], and Metaboanalyst [35–37] are the ones used by our group. For most applications, raw MS data are converted to the mzML [38] or mzXML [39] standard types before processing, using ProteoWizard MSConvert [40].

## 3. Materials and Equipment

### 3.1. Reagents

All solvents and reagents for mass spectrometry must be of LC/MS grade. Acetonitrile, methanol, water, isopropanol, ammonium acetate, ammonium formate, formic acid and acetic acid (LC/MS grade) are from Fisher Scientific. L-tryptophan (indole-$d_5$) and L-valine ($d_8$) are from Cambridge Isotope Laboratories. N,N-dimethyl-$d_6$ glycine HCl is from CDN Isotopes. In addition, 0.5 M EDTA solution, pH 8.0 is from Millipore. All other reagents are from Sigma-Aldrich. Full reagent details are given as Supplementary Information.

### 3.2. Experimental Models

The protocol steps regarding animal metabolomics were optimized using healthy 13-week-old C57BL/6J male mice. Additional details regarding the entire study can be found in Marques et al. [19]. This protocol, though optimized using mouse tissue, can be easily transferred to other species.

Primary cell cultures are also suitable for metabolomics assays. In our group, we have used embryonic chicken neurons to probe the effect of drugs on the metabolome profile. Complete details can be found in the literature [19,41].

All animal procedures were performed in accordance with the EU Directive 2010/63/EU for animal experiments and the Portuguese Law for Animal Care for animal use in research (DL 1/2019), in compliance with the authors' institutional review board directives (Instituto Superior Técnico, Universidade de Lisboa) and the ARRIVE 2.0 guidelines for planning, conducting, and reporting animal research [42].

### 3.3. Equipment

- UHPLC system composed of an Elute UHPLC HPG 1300 pump with two pairs of serial-coupled, individually controlled linear drive pump heads, an Elute autosampler and an Elute CSV column oven preheater, equipped with C18 reverse phase and a HILIC column, with pre-columns, and coupled to an Impact II QqTOF mass spectrometer with an electrospray ion source (UHR-ESI-QqTOF, Bruker Daltonics GmbH & Co., Bremmen, Germany). Further details are available and discussed in the Supplementary Materials;
- Vials with caps and inserts;
- Polypropilene microcentrifuge tubes, 1.5 mL and 2 mL;
- 0.22 μm polyethersulfone (PES) membrane (Branchia, cat. # SFPE-22E-050);
- Ultrasonic bath (Bandelin Electronic, RK 52);
- Potter–Elvehjem homogenizer (Sigma (St. Louis, MO, USA), cat. # P7734);
- Micro-pestles for 1.5 and 2.0 mL tubes (Carl Roth, cat. # CXH7.1);

- SPECTROstar BMG Labtech equipped with a multiwell plate reader and a low-volume microspot plate (BMG Labtech, SpectrostarNano).

### 3.4. Software

- Data Analysis v4.1-4.5 (Bruker (Billerica, MA, USA));
- MARS software v3.32 (BMG Labtech);
- ProteoWizard MSConvert v3.0 (https://proteowizard.sourceforge.io/, accessed on 3 February 2022) [40];
- XCMS v3.7.1 (https://xcmsonline.scripps.edu/, accessed on 1 October 2021) [32];
- Metaboanalyst v5.0 (https://www.metaboanalyst.ca/, accessed on 4 October 2022) [35,36];
- MZMine v3 (http://mzmine.github.io/, accessed on 4 October 2022) [30,31].

### 3.5. Reagent and Sample Setup

#### 3.5.1. Samples

Samples should be prepared fresh before analysis or stored at $-80\ ^{\circ}$C until analysis.

#### 3.5.2. Quality Control (QC)

A 25 µM solution containing quercetin, L-tryptophan (indole-$d_5$), L-valine ($d_8$), sulfolene ($d_4$), and *N*,*N*-dimethyl-$d_6$ glycine HCl was prepared and used as quality control.

#### 3.5.3. Mobile Phase Solutions

Reverse-phase chromatography:

- Mobile phase A1 is 0.1% (*vol/vol*) formic acid in water; this is prepared by adding 1.0 mL of formic acid to 1 L of LC-MS-grade water and mixing thoroughly.
- Mobile phase B1 is 0.1% (*vol/vol*) formic acid in acetonitrile; this is prepared by adding 1.0 mL of formic acid to 1 L of LC-MS-grade acetonitrile and mixing thoroughly.

Hydrophilic interaction chromatography:

- Mobile phase A2 is 10 mM ammonium acetate in water, with 0.1% (*vol/vol*) acetic acid. This is prepared by dissolving 0.7708 g of ammonium acetate in 1 L of LC-MS-grade water and adding 1.0 mL of acetic acid.
- Mobile phase B2 is 10 mM ammonium acetate in acetonitrile containing 2% (*vol/vol*) water and 0.1% (*vol/vol*) acetic acid). This is prepared by (i) dissolving 0.7708 g of ammonium acetate in 20 mL of LC-MS grade water, (ii) diluting this with 980 mL of warm LC-MS-grade acetonitrile, and (iii) adding 1 mL of acetic acid. Note: Mobile phase B2 must be prepared by adding warm acetonitrile (ca. 37 $^{\circ}$C) or ammonium acetate will precipitate. Upon proper dissolution, no temperature-dependent precipitation has been observed.

#### 3.5.4. Sodium Formate/Acetate Calibrant Solution

Instrument calibration is performed routinely with a 1 mM 1:1 solution of formate and acetate clusters in 50/50 isopropanol:water with 0.2% (*vol/vol*) mixture of 3:1 (*vol/vol*) acetic acid:formic acid. Detailed information is presented in Table S1. This solution can be used for up to 4 weeks and stored at room temperature.

### 3.6. Equipment Setup

#### 3.6.1. UHPLC Instrument Setup

Two chromatographic methods are used: reverse-phase chromatography and hydrophilic interaction chromatography, given in further detail in Tables S2 and S3.

For RP chromatography, a Luna C18(2)-HST column is used. The injection volume is set to 5 µL and the autosampler temperature is set to 8 $^{\circ}$C. The column oven is set at a constant temperature of 40 $^{\circ}$C, and a 250 µL/min flow rate gradient elution is used, with

mobile phases A1 and B1, as follows: 0 min, 0% B1; 0.5 min, 0%B1; 1.5 min, 20% B1; 4.0 min, 60% B1; 6.0 min, 100% B1; 9.0 min, 100% B1; 10.0 min, 0% B1; 15.5 min, 0% B1.

For HILIC, an XBridge BEH Amide XP column is used. Injection volume is set to 5 μL and the autosampler temperature is set to 8 °C. Column oven is set at a constant temperature of 40 °C, and a 250 μL/min flow rate gradient elution is used, with mobile phases A2 and B2, as follows: 0 min, 90% B2; 2.0 min, 0%B2; 6.0 min, 70% B2; 9.0 min, 30% B2; 13.0 min, 30% B2; 18.0 min, 90% B1; 22.0 min, 90% B2.

The injection volume is highly dependent on the sample composition. We recommend performing a test run using a sample and injecting 5 μL. Chromatographic results (total ion chromatogram and base peak chromatogram) can be analysed to decide whether that is an adequate injection volume or if it should be increased or lowered.

### 3.6.2. Mass Spectrometer Setup

The mass accuracy of QqTOF is guaranteed through the regular calibration of the mass spectrometer performed using the calibration solution described above in Section 3.5.4.

High-resolution mass spectra are acquired in both ionization modes, with the following acquisition parameters: capillary voltage, 4.5 kV (electrospray positive mode) or 3 kV (electrospray negative mode); end plate offset, 500 V; nebulizer, 2.0 bar; dry gas ($N_2$) flow, 8.0 L/min; dry heater temperature, 220 °C. The tune parameters were set according as follows: transfer funnel 1/2 RF power (150/200 Vpp), hexapole RF power (50 Vpp), ion energy (4.0 eV), low mass (90 *m/z*), collision energy (7.0 eV), collision RF power (650 Vpp), transfer time (80 μs), pre-pulse storage (5 μs). Spectra acquisition was performed with an absolute threshold of 25 counts per 1000. For auto-MS/MS mode, spectra are acquired with a threshold of 20 counts per 1000, cycle time of 3.0 s with exclusion after 3 spectra and release after 1.00 min. All acquisitions are performed with an *m/z* range from 70 to 1000 and with a 3 Hz spectra rate. Three full scans and 1 auto MS/MS scan are performed for each sample using both positive and negative ionization modes, for both RP and HILIC separation modes.

Note: Data acquisition rate (spectra rate) is highly dependent on the equipment used and on the samples. Higher scan rates, about 10 Hz, can be used to potentially acquire more data, but at the cost of reduced ion intensities, which can be partially (but not totally) compensated by a higher injection volume.

## 4. Expected Results and Data Analysis

In this protocol, samples from different origins are prepared for mass spectrometry-based metabolomics studies. Each sample is analysed using two chromatographic approaches—an RP approach and a HILIC approach—with two ionization modes (positive and negative), generating full MS and MS/MS data on all modes. A number of possible problems during sample preparation and analysis are troubleshooted in Table S4.

Following an open-source software approach to data treatment, we convert the acquired raw data to mzXML [39] format, with no filters or thresholds applied, using ProteoWizard MSConvert [40]. These can then be imported into XCMS or Metaboanalyst servers, or to local MZMine installations, for processing. Regardless of the software, certain data processing steps are mandatory. Upon raw data import, a mass detection step identifies the ion *m/z* values in all data files above a given intensity and allows the construction of ion chromatograms, one per *m/z* value identified per raw data file. Following various chromatogram smoothing and resolving/deconvoluting steps, isotopic filters must be applied to remove isotopologues. From this step on, peaks are aligned across the various samples, and an annotation step tries to attribute a correct identification to each identified ion.

XCMS is one of the most user-friendly software packages for metabolomics data analysis. It can be used through the online server hosted at The Scripps Research Institute (available at xcmsonline.scripps.edu) or through local implementations, for example, via Bioconductor [43]. We detail below the overall approach for untargeted metabolomics data analysis using the online XCMS server.

The first step is to upload the mzXML files to the server, following an acquisition mode-centric logic. For each analysed sample (test or control), data files from each acquisition mode (RP positive, RP negative, HILIC positive, HILIC negative) must be uploaded to individual datasets, with clear names.

Following this, data analysis follows a pairwise comparison approach. Data from each tested condition, in each acquisition mode, are compared to data from a control dataset, analysed in the same acquisition mode. For this, a pairwise job can be created by selecting the sample data as the first dataset and the control data as the second dataset. Parameters must be selected according to the type of instrument and instrumental parameters. In our case, using a Bruker Impact II TOF spectrometer, we found that the default "UPLC/Bruker Q-TOF pos" was a good starting point. When editing each method, particular attention must be given to (i) the polarity parameter (General tab), (ii) the *m/z* tolerance for feature identification between different scans (parameter ppm in the Feature detection tab), and (iii) the retention time deviation (parameter bw in the Alignment tab). Additionally, the sample biological source must be properly chosen in the Identification tab.

In XCMS, ion chromatograms are only extracted when peaks are present above a given threshold abundance. By default, this value is set at 100, but can be increased in the General tab of the method. These ion chromatograms undergo integration to determine peak areas, which will later be used to compute fold changes between samples.

As an example, Figure 6 shows the output obtained from running a pairwise job on positive ionization RP data from mice brain tissues, comparing montelukast-treated animals with control animals [19]. XCMS' pairwise jobs encompass all steps mentioned above, and, by performing a pathway enrichment, generate system biology-level results, identifying biological pathways significantly altered between test and control.

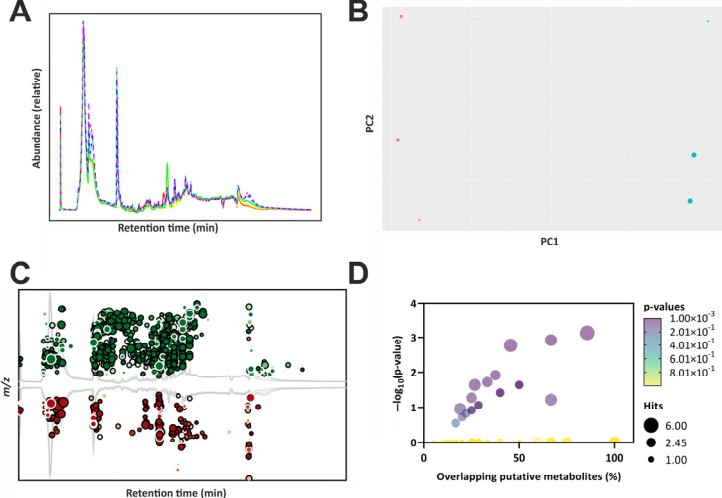

**Figure 6.** Sample XCMS pairwise output. An XCMS pairwise job compares test samples with controls analysed in the same chromatographic conditions and the same ionization mode. The results obtained by analysing brain samples from montelukast-treated mice and control animals (RP, positive mode) are shown. Full experimental details are available in the original publication [19]. (**A**) XCMS-generated total ion chromatograms. (**B**) Principal component analysis of the six full scan analyses performed grouping test animals (red dots) and control animals (blue dots). While not informative in this case, for larger samples and/or a larger number of conditions, this analysis offers a clear identification of whether different samples are significantly different or not. (**C**) Cloud plot representation of the identified ions comparing their abundance in test animals and controls. Green circles correspond to upregulated features and red circles to downregulated features. Circle size is proportional to fold change, and circle shading is proportional to the significance of that change (darker shades correspond to a smaller *p* value). (**D**) Cloud plot of the dysregulated pathways, obtained via pathway enrichment with the identified dysregulated metabolites.

XCMS also outputs a diffreport spreadsheet file that can be used to produce a Volcano plot (Figure 7). In a typical Volcano plot of metabolomics data, the *xx* axis represents the fold-change of the relative abundance of each metabolite in the sample vs. control (in a log2 scale) and the y axis represents the statistical significance of the ratio fold-change of each metabolite (in a log10 scale). Statistically increased or decreased metabolites will appear in the upper right and upper left regions of the plot, respectively.

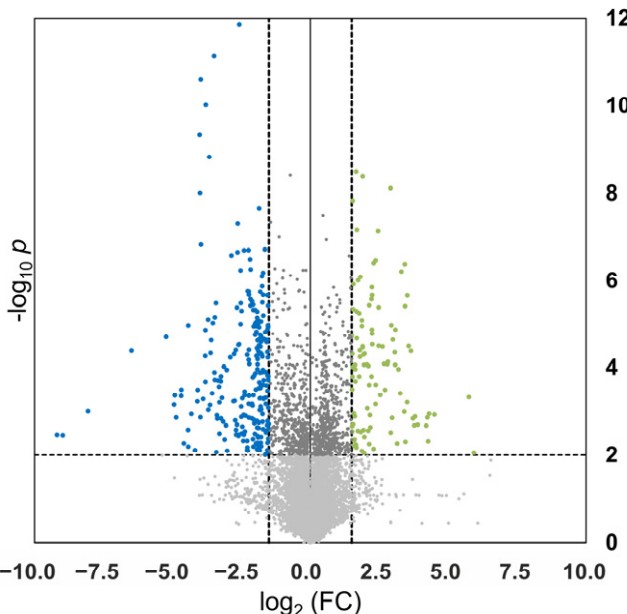

**Figure 7.** Representative volcano plot of metabolomics data. The *xx* axis represents the $\log_2$ value of the fold-change of relative abundances of each metabolite in the test v. control samples, and the *yy* axis represents the statistical significance (*p* value) of the calculated fold-change ratios. The horizontal dashed line (at *p* = 0.01) separates the significant values (above the line) from non-significant values (below the line). Metabolites with higher abundance in the test samples are found to the right of the *yy* axis (FC > 1) and those with higher abundance in the control samples are found to the left of the *yy* axis (FC < 1). The two vertical dashed lines, at $\log_2$ (FC) = 1.5 and −1.5, together with the horizontal dashed line at *p* = 0.01 delimit two regions of the plot, where metabolites were found significantly increased (green dots) or significantly decreased (blue dots) in the test samples (relative to control samples). Each dot corresponds to a different metabolite.

Following the four required pairwise jobs (1 per each acquisition mode), a multimodal job is required to merge and align all data from the individual pairwise jobs. For these, a multi-modal job is created by selecting the 4 pairwise associated jobs; the biological source must be confirmed. In this case, the brain metabolome of montelukast-treated animals exhibited many altered pathways (one per circle on the cloud plot on the left), of which the 10 pathways most significantly altered are given in the table on the right [19] (Figure 8). For example, for the top pathway, the tRNA charging pathway, 14 altered metabolites were identified, and these 14 metabolites comprise 66.7%, or 2/3, of the whole pathway. Considering the prostaglandin pathway, all metabolites (100%) were found to be altered in the treated animals vs. controls.

While a pairwise job will only reveal alterations from metabolites identified in one of the four experimental modes, multimodal jobs will group together all jobs, which allows (i) for the merging of metabolites identified in more than one experimental mode and (ii) for a deeper pathway enrichment, as most metabolites are not observed in all experimental modes.

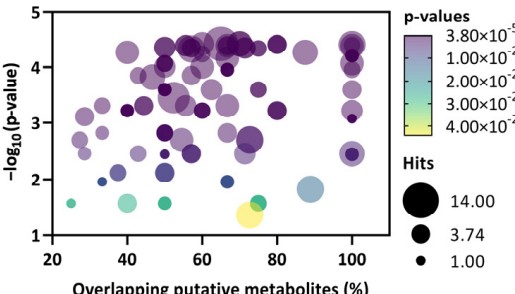

| Metabolic Pathways (Top 10) | Hits | Overlapping putative metabolites | p-value |
|---|---|---|---|
| tRNA charging pathway | 14 | 66.70% | $3.8 \times 10^{-5}$ |
| Biosynthesis of corticosteroids | 7 | 70.00% | $3.8 \times 10^{-5}$ |
| Purine and pyrimidine metabolism | 13 | 65.00% | $3.8 \times 10^{-5}$ |
| Salvage pathways of pyrimidine ribonucleotides | 6 | 60.00% | $3.8 \times 10^{-5}$ |
| Arginine biosynthesis IV | 7 | 70.00% | $3.8 \times 10^{-5}$ |
| Asparagine biosynthesis I | 4 | 80.00% | $3.9 \times 10^{-5}$ |
| Glutathione biosynthesis | 4 | 80.00% | $3.9 \times 10^{-5}$ |
| Spermine biosynthesis II | 4 | 80.00% | $3.9 \times 10^{-5}$ |
| Biosynthesis of prostaglandins | 8 | 100.00% | $3.9 \times 10^{-5}$ |
| Spermine biosynthesis | 3 | 100.00% | $4.0 \times 10^{-5}$ |

**Figure 8.** Mouse brain metabolic pathways significantly altered upon MTK administration. Cloud plot (**left**) shows the significantly altered pathways and their metabolite overlap (fraction of identified altered metabolites relative to the total number of metabolites that constitute the pathway), as well as the number of hits per pathway. On the (**right**), the top 10 altered pathways are listed. Statistical significance: $p < 0.0001$. Full experimental details are given in the original reference. Reprinted from Marques et al. [19], with permission from Elsevier.

Handling such large datasets to retrieve significant information must be accompanied by proper statistics testing, starting at the pairwise job level. The online XCMS server uses a default "Unpaired parametric *t*-test (Welch *t*-test)", but others can be chosen in the Statistic tab when editing the XCMS method. The Welch *t*-test, an adaptation of Student's *t*-test, enables the testing of the null hypothesis that two populations have equal means, and is more reliable that the original Student's version when the two samples have unequal variances and unequal sample sizes. This test is also commonly referred to as Welch's unequal variances *t*-test, to emphasize the different variance approach. It must be noted that this test still assumes population normality.

A paired parametric (assuming normality) *t*-test is also available for selection. The choice of whether a paired or unpaired test should be used depends on the experimental design. For example, if each individual animal, before treatment, is uniquely matched to the same individual after treatment, a paired analysis can be followed. If a set of animals are randomly assigned to two different groups, an unpaired analysis strategy must be employed.

Two other non-parametric tests are available for selection in XCMS—the unpaired Mann–Whitney test and the paired Wilcoxon signed rank test. Non-parametric tests are relevant when samples do not follow a distribution or when distribution parameters are unknown, or assumptions are violated. Preliminary testing can help decide whether parametric or non-parametric tests should be used [44]. A comprehensive review of this subject can be found in Vinaixa et al. [44].

Another important aspect of untargeted metabolomics is the assignment of a metabolite identification to each *m/z* feature. Online XCMS uses METLIN for such annotation, with over 7000 metabolomes available coming from different species, cell lines, strains, etc. If the relevant metabolome is not available, an alternative hypothesis for data processing is using MZMine, which allows you to use in-house build metabolome databases from as little as metabolite ID and *m/z*. For a more detailed explanation on how to use MZMine, readers are referred to the documentation or a worked-out example [45].

## 5. Conclusions

These protocols correspond to a generic untargeted approach to mass spectrometry-based metabolomics assessment and can be applied to different types of biological samples. While the specialized literature will describe large numbers of customized extraction protocols for each type of biological sample, often optimized for each species of bacteria or each cell type, these protocols were developed for a generic and fast application in the lab, towards an initial untargeted approach.

The primary goal is to allow a fast analysis of a large number of samples, without any preliminary metabolic/metabolomics assumptions applied, and to mine the collected data in order to identify dysregulated biochemical pathways in test samples (relative to control).

This untargeted approach can be followed by targeted analysis, focusing on particular pathways, and for which optimized sample preparation protocols might be desirable.

**Supplementary Materials:** The following supporting information can be downloaded at: https://www.mdpi.com/article/10.3390/separations10050314/s1, Detailed step-by-step protocol; Table S1: Preparation of calibrant solution used for high-resolution mass spectrometry internal and external calibration; Table S2: Reversed-phase UHPLC-MS gradient; Table S3: HILIC UHPLC-MS gradient; Table S4: Troubleshooting.

**Author Contributions:** Conceptualization, C.F.M. and G.C.J.; methodology, C.F.M. and G.C.J.; formal analysis, C.F.M. and G.C.J.; investigation, C.F.M.; resources, G.C.J.; writing—original draft preparation, C.F.M.; writing—review and editing, C.F.M. and G.C.J.; visualization, C.F.M.; supervision, G.C.J.; funding acquisition, G.C.J. All authors have read and agreed to the published version of the manuscript.

**Funding:** This research was funded by Fundação para a Ciência e a Tecnologia, grants number PTDC/QUI-QAN/32242/2017, RNEM-LISBOA-01-0145-FEDER-022125 (RNEM—Portuguese Mass Spectrometry Network), PD/BD/143128/2019 and COVID/BD/152559/2022 (CFM). Centro de Química Estrutural is a Research Unit funded by FCT through projects UIDB/00100/2020 and UIDP/00100/2020. Institute of Molecular Sciences is an Associate Laboratory funded by FCT through project LA/P/0056/2020.

**Data Availability Statement:** No new data were created or analysed in this study. Data sharing is not applicable to this article.

**Acknowledgments:** The authors would like to thank Maria da Conceição Oliveira, for the long and fruitful discussions on chromatography and spectrometry theory and practice applied to metabolomics, and to dedicate her this work, on her retirement, for the encouragement in implementing an omics line of work at the RNEM Técnico's node, which she has led since its origin in 2005.

**Conflicts of Interest:** The authors declare no conflict of interest.

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
