# Peer review of "An Optimised MS-Based Versatile Untargeted Metabolomics Protocol"

_separations, doi:10.3390/separations10050314_

Round 1
Reviewer 1 Report
The presented manuscript shows valuble protocols for LC-MS based metabolomics. I have only one suggestion. The Fig. 6 B shows the PCA graph for two samples in one group. PCA method is not adequate for small experimental groups. I think, that you sould replaces PCA by volcano-plot or fold change analysis.
Reviewer 2 Report
The manuscript titled “An optimised MS-based versatile untargeted metabolomics protocol” is written in detail with respect to the different set of protocols that can be applied to various types of samples, including prokaryotic and eukaryotic cells, as well as animal and human samples by following a single extraction step, and analyzing using different chromatographic conditions coupled to high-resolution mass spectrometry.
The figures were well designed to understand and easily readable to the readers.
Definitely this optimized protocol will help the readers for exploring and identifying the biological pathways of potential relevance to the biological conditions assayed.
And these protocols will help the researchers who are performing the untargeted metabolomic studies of patients’ samples that has the potential to unravel novel candidate biomarkers that can ultimately translate into a faster disease diagnosis and management. Although targeted approach should be used for absolute quantification of such discovered biomarkers and in validating those biomarkers for disease diagnosis.
The author would have explored a little bit more on the bioinformatics part for this optimized MS-based untargeted Metabolomics approach.
Overall, the manuscript can be accepted with minor revisions
- by adding more references in the experimental design section.
- Grammar mistakes in some places are found and should be corrected.
Reviewer 3 Report
The manuscript entitled “An optimised MS-based versatile untargeted metabolomics protocol” describes a set of sample preparation protocols combined with a universal LS-MS analysis protocol for untargeted screening of different metabolite changes related to infection, disease or intake of pharmaceuticals.
The protocols are clearly presented and can be implemented in the working process of different laboratories for the broad area of research purposes.
There are, however, some questions arisen from the text of the manuscript:
1) What are the key differences of the suggested protocol from those already published in literature?
2) How exactly relative quantification of metabolites is performed? Short description “…using peak areas from total ion current for statistical analysis” is given, which is misleading because some metabolites are on the low concentrations and can not be observed directly in Total Ion Current chromatograms. How ionization problems are dealt in case of severe and (simultaneously) different matrix effects in the studied samples?
